# Contribution of Nucleotide-Binding Oligomerization Domain-like (NOD) Receptors to the Immune and Metabolic Health

**DOI:** 10.3390/biomedicines12020341

**Published:** 2024-02-01

**Authors:** César Jeri Apaza, Marisol Días, Aurora García Tejedor, Lisardo Boscá, José Moisés Laparra Llopis

**Affiliations:** 1Molecular Immunonutrition Group, Madrid Institute for Advanced Studies in Food (IMDEA Food), Ctra Cantoblanco, 8, 28049 Madrid, Spain; immuno.cesar@imdea.org; 2Center of Biological Enginneering (CEB), Iberian Nantotechnology Laboratory (INL), University of Minho, 4715-330 Braga, Portugal; marisol.dias@inl.int; 3Bioactivity and Nutritional Immunology Group (BIOINUT), Faculty of Health Sciences, Universidad Internacional de Valencia (VIU), Pintor Sorolla 21, 46002 Valencia, Spain; agarciate@universidadviu.com; 4Instituto de Investigaciones Biomédicas Alberto Sols-Morreale (CSIC-UAM), Arturo Duperier 4, 28029 Madrid, Spain; lbosca@iib.uam.es; 5Centro de Investigación Biomédica en Red en Enfermedades Cardiovasculares (CIBERCV), Melchor Fernández Almagro 6, 28029 Madrid, Spain

**Keywords:** NOD1/2, Toll-like receptors, immunonutrition, innate immunity, cancer

## Abstract

Nucleotide-binding oligomerization domain-like (NOD) receptors rely on the interface between immunity and metabolism. Dietary factors constitute critical players in the activation of innate immunity and modulation of the gut microbiota. The latter have been involved in worsening or improving the control and promotion of diseases such as obesity, type 2 diabetes, metabolic syndrome, diseases known as non-communicable metabolic diseases (NCDs), and the risk of developing cancer. Intracellular NODs play key coordinated actions with innate immune ‘Toll-like’ receptors leading to a diverse array of gene expressions that initiate inflammatory and immune responses. There has been an improvement in the understanding of the molecular and genetic implications of these receptors in, among others, such aspects as resting energy expenditure, insulin resistance, and cell proliferation. Genetic factors and polymorphisms of the receptors are determinants of the risk and severity of NCDs and cancer, and it is conceivable that dietary factors may have significant differential consequences depending on them. Host factors are difficult to influence, while environmental factors are predominant and approachable with a preventive and/or therapeutic intention in obesity, T2D, and cancer. However, beyond the recognition of the activation of NODs by peptidoglycan as its prototypical agonist, the underlying molecular response(s) and its consequences on these diseases remain ill-defined. Metabolic (re)programming is a hallmark of NCDs and cancer in which nutritional strategies might play a key role in preventing the unprecedented expansion of these diseases. A better understanding of the participation and effects of immunonutritional dietary ingredients can boost integrative knowledge fostering interdisciplinary science between nutritional precision and personalized medicine against cancer. This review summarizes the current evidence concerning the relationship(s) and consequences of NODs on immune and metabolic health.

## 1. Introduction

Today, more than 1600 million people (aged 15 years and more) worldwide are overweight or obese and, according to the World Health Organization (WHO), this number will increase to 2300 million in 2050. Worldwide, hyperglycemia kills some 3.4 million people a year. In the EU, approximately 60% of adults and 20% of children of school age are overweight or obese (WHO, (https://www.who.int/, accessed on 1 November 2023). In the USA, the situation is very similar where being overweight and obese occurs in the population in an approximate proportion of 3/4 of adults (>20 years) and 1/5 of children and adolescents (2–19 years) [1]. The WHO forecasts that deaths from type 2 diabetes (T2D) will double between 2005 and 2030. In these contexts, the metabolic (re)programming of certain subphenotypes of immune system components, which are key to inducing obesity (i.e., innate and macrophage lymphoid cells), is a distinctive stamp in alterations to the homeostasis of nutrients in which nutritional strategies can play a key role. Data from epidemiological studies reveal that obesity and T2D significantly contribute to increasing the risk for a number of cancer types [2]. Overall, a reduction can be seen when considering dietary risks as the origin of carcinogenic processes [3]. However, it is worth noting the increasing trend in the incidence of cancer if we consider factors such as weight gain and metabolic pathologies such as diabetes [4]. Cancer represents a scourge of today’s societies [1]; there were an estimated 18.1 million cancer cases around the world in 2020. Of these, 9.3 million cases were in men and 8.8 million in women.

The role of food ingredients, beyond their nutritional value, has been a subject of an open debate over the past few decades. In this sense, the European Food Safety Authority (EFSA) has issued specific guidelines referring to health claims pertaining to various foods/food constituents [5] as well as to intestinal immunity and defense against pathogens [6]. These guidelines stress the need for establishing a cause–effect relationship and definition of specific functions of the immune system to be improved by foods/food constituents. Nowadays, most nutritional interventions possess a clear observational perspective rather than a hypothesis-driven point of view. Despite significant advances in the understanding of metabolic diseases such as obesity, type 2 diabetes, metabolic syndrome, and cancer, dietary recommendations have remained general and, at this point, apply to all patients regardless of the disease features. In this sense, “Precision nutrition” has emerged as a discipline with the potential to provide significant contributions to successfully implement personalized medicine [7,8]. Understanding how a host’s intrinsic features determine the metabolic variability between individuals is used to tailor specific nutritional strategies to manage diseases.

Previous research has largely focused on total calorie intake and an adequate nutrient intake so as to develop proper immune system response(s) [9]. NCDs share common features, such as liver dysfunction and tissue inflammation, where the monocyte/macrophage population has multifaceted effects [10,11]. Recent findings also provided new insights into how innate and adaptive lymphocytes operate sequentially and in distinct ways during normal development to establish steady-state commensalism and lipid homeostasis [12]. Studies on the intricate relationship between the maturation of the intestinal immune system and the induction of obesity revealed that innate lymphoid cells (group 2)—ILC2s—are determinants in the induction of diet-induced obesity [13]. Later research demonstrated that liver macrophages are determinants in the diet-regulated control of hepatic fat accumulation [14]. In this sense, recent data evidence that dietary nutrients able to promote beneficial immune response(s)—immunonutritional agonists—can be even more important determinants of intestinal, liver, metabolic, and cardiovascular health [15]. These effects appear to be derived from a selective functional differentiation of the monocyte/macrophage population towards an M1-like phenotype [15]. In addition, recent research also associated the expansion of ILC2 and ILC3 populations with a reduction in the accumulation of fats into the liver [16].

Collectively, a critical overview of the scientific literature indicates that NCDs and cancer require effective nutritional intervention strategies. The latter is preferably based on immunonutritional agonists enabling the oriented expansion and activity of innate immune populations. The modulation of immune responses via innate immune pattern recognition receptors by immunonutritional agonists are strategies that could have determinant influences on promoting the metabolic and immune health of the gut–liver axis. In particular, nucleotide-binding oligomerization domain-like (NOD) receptors rely on the interface between immunity and metabolism. Thus, the integration of signaling molecule regulators of these receptors constitutes a promising strategy for approaching, with a preventive and/or therapeutic intention, immunometabolic diseases such as obesity, T2D and cancer. A better understanding of these processes can help to integrate the potential use of NODs in targeted precision nutrition strategies as coadjuvant to the classical pharmacological treatments (Figure 1).

This review summarizes the current evidence concerning the relationship(s) and consequences of NODs on immune and metabolic health. An extensive search was carried out to identify as many studies as possible relevant to NOD receptors. Particular efforts were devoted to compiling human intervention studies as preclinical studies are enormously informative, but the effects may not be recapitulated in human trials. All relevant studies are classified according to the reported biomarkers associated with NCDs and cancer, and to affect the metabolic and innate immune function.

## 2. Current Challenges in the Nutritional Approach to NODs in NCDs and Cancer

Over the last decade, there has been a significant implementation of nutritional approaches for NCD and cancer patients [7,17,18]. Only a few of the main drivers responsible for consistent clinical outcomes in diabetes (i.e., type 1, 2, or prediabetes) have been identified as nutrient-targeted processes [18]. Nutritional obesity management approaches the modulation of inflammation to improve insulin sensitivity, and consequently fat accumulation [17]. In 2022, a panel of experts on the nutritional approach for cancer patients in Spain agreed upon the extended incidence of malnutrition in cancer patients [19]. In general terms, nutritional approaches rely on the energetic and biological value of nutrients. The study of interindividual dissimilarities based on exposome heterogeneity (i.e., genetic/epigenetic, lifestyle, microbiome and behavioral/psychological features) significantly contributed to improving disease management [7]. Eminently, these studies attempted to condition the intensity and severity of potential interactions to augment the magnitude of the effects attributable to nutrient utilization. There exists a link between immunocompetent cells (i.e., the monocyte/macrophage population and ILCs) and functions and the development of NCDs and cancer [12,13,14]. However, addressing these processes to modulate the oriented activation and function of these immune effectors by foods/food constituents remains poorly exploited.

Preclinical studies have reported opposing results concerning the implication and role of NOD1 in the promotion and severity of obesity [20,21]. These effects vary from the protective role of NOD1 deficiency against obesity-induced inflammation and insulin resistance [20], to a deleterious effect accelerating diet-induced obesity and liver steatosis in HFD-fed mice [21]. NCDs are characterized by low-grade inflammation. In this context, peripheral monocytes and macrophages derived from these appear as central players in the progression of liver dysfunction, worsening or improving the disease. This inducible population of innate immune effectors acquires a specific phenotype due to the integration of signals (i.e., immune and metabolic) originating (i.e., tissular and systemic) within the gut–liver axis. In NCDs, the role played by M1-like macrophages under chronic/sterile inflammation allows us to consider a situation where NOD1 is subjected to inhibitory regulation. This assumption is based on reduced macrophage apoptosis derived from NOD1 deficiency [22], and the contrasting apoptosis-mediated clearance of macrophages from resolving inflammation [23].

Prior research has shown the key role of intestinal innate immunity in determining lipid homeostasis and gut microbiota composition [12,15]. These studies revealed deep links between the immune system and body organ metabolism. While ILC3s have been shown to exhibit a clear role in the establishment of a tolerable commensal state and influence lipid homeostasis [12], ILC2s appear as essential determinants of fat absorption [13]. In addition, taking advantage of preclinical models—deficient in adaptive immune effectors, Rag2^−/−^ mice and ILCs, Rag2^−/−^IL2^−/−^ mice—it has been shown that the promotion of an M1-like phenotype (CD68^+^F4/80^+^) of the monocyte/macrophage population was associated with the expansion of ILC2/3 (i.e., CD117^+^KLRG^+^ group ILC2s and CD56^+^CD117^+^Nkp46^+^ group ILC3s) precursors that benefit the control of HFD-induced obesity [16]. The latter appeared as a consequence of the adaptations of lipid homeostasis, which could not be associated with changes in the gut microbiota. Altogether, these results are concordant with the previously reported role as responsible for NOD2 sensing to selectively activate inflammatory cytokine production from ILC2/3s [24].

Overall, there are substantial knowledge gaps in relation to the integration of NODs into nutrition strategies, eco-nutritional conditions, immunity, and the specificity of the effects (Figure 1): (i) understanding food-derived ingredients’ influence on NODs at the systems’ biology level allowing for the comparison of different sources, (ii) producing new knowledge on stimulatory and inhibitory effects, as well as (iii) integrating information as complementary to the classical pharmacological approach. Despite the well-known role of NOD receptor proteins in the interplay between the microbiota and gastrointestinal immune adaptations [25], their regulation by foods/food constituents in NCDs is less explored. Most available information identifies the influence of phytochemicals and extracts of aromatic and medicinal plants [25,26] as signaling molecule modulators. These compounds may display the potential to access the cellular plasmatic membrane, thus impairing TLR4 relocation and distribution to lipid rafts [27]. However, the impossibility to control the effective concentration of these compounds to modulate signaling in an oriented fashion by directly interacting with the receptors limits their use with a clinical preventive and/or therapeutic intention.

## 3. NOD/NLR Signaling: Implications for NCDs and Cancer

Elegant reviews have compiled information concerning the structure and gene encoding for NODs [28,29]. NOD1 and NOD2 are located in the cytoplasm and are composed of an LRR ligand-binding domain, an oligomerization domain with NACHT homology, and a caspase recruitment domain (CARD) that transmits the signal. Despite these significant advances and improvements in the understanding of the genetic basis for these receptors, their importance in NCDs and cancer could be underscored by the fact that single-nucleotide polymorphisms (SNPs) occurring in different subpopulations of patients have been barely elucidated [30,31]

NOD1/2 and NOD-like receptors (NLRs) belong to the superclass of pattern recognition receptors (i.e., PAMPs, DAMPs), among which are also the innate immune ‘Toll-like’ receptors (TLRs). Pattern recognition receptors play key roles in determining the function and activity of innate immune effectors. Both NODs/NLRs and TLRs are well-recognized mediators of the immune and metabolic imbalances occurring in NCDs and cancer [32,33,34]. It is well known that microbial peptidoglycans—γ-D-glutamyl-meso-diaminopimelic acid and muramyl dipeptide—act as the prototypical agonist of NODs, and despite the known and described functions as microbial sensors, they also affect the development of extraintestinal diseases and cancer [35]. For example, NOD1/2 activation and deficiency as well as the intake of a diet rich in lipids and certain signals of cellular damage are closely related to cell proliferation and the response(s) to chemotherapy in hepatocellular carcinoma (HCC) [32]. The balance between the anti/pro-tumor consequences from NOD signaling appears to depend on several factors (i.e., the type and stage of the tumor, the microenvironment, and the interactions between the cells of the immune system) [36]. The convergence between NODs/NLRs and TLRs may constitute the way to integrate the different stimuli determining the underlying signaling and the receptors’ contribution to disease development.

Recent work identified the participation of Rho GTPases as part of the molecular signaling associated with NODs [37]. In humans and mice, the widespread expression of Rho kinases, which represent target molecules in regulating metabolic function and energy storage, is known [38]. A clear example of the importance of NCDs is that molecules such as Rac1 play important positive roles in insulin-stimulated glucose uptake. The latter effect constitutes a possible link between obesity and T2D. Rac1 also represents a regulator of cell migration and a potential target for cancer therapy and contributes to the maximal activation of STAT3 in IFN-γ stimulation. Rho kinases are also relevant players in cancer and neurodegeneration [39]. Not surprisingly, Rho kinases, also exert various activities to effectively modulate the immune system and have been identified as potential therapeutic targets in cancer immunotherapy [40]. Protein kinase inhibitors can be found in several foods where bioactive compounds displaying inhibitory activity occur naturally [39]. Unfortunately, the inhibitory role appears unspecific and reversible, as in the case of polyphenols. In addition, some of these activities have their origin in both bacterial production and their metabolic capacities [41]. Notwithstanding, recent work has shown the potential of serine-type protease inhibitors to up-regulate the Rho GDP-dissociation inhibitor in macrophages via interaction with TLR4 (MyD88-independent) signaling [42]. This effect was accompanied by the downregulatory effect of several glycolytic mediators, thus allowing us to hypothesize a negative regulatory effect on NOD signaling. Collectively, these observations suggest that imbalances of cellular metabolic homeostasis are sensed by NOD1/2, thereby contributing to the early glycolytic reprogramming of human monocyte-derived macrophages [33]. The implication of NOD2 in the production of type I interferons, associated with the interconnected signaling with TLRs, could be responsible, at least in part, for boosting the loss of regulatory capacity on the proinflammatory processes in NCDs.

A direct consequence of the metabolic imbalances occurring during NCDs and cancer is increased stress on the endoplasmic reticulum [43]. This organelle plays a critical role in the development of NCDs and cancer since the maintenance of its homeostasis is essential in the regulation of TLR4 expression. In this context, the capacity of NOD1/2 for sensing ER stress may also be of relevance for these pathologies in which the receptors could significantly influence the organelle’s function. ER stress has been identified as a central feature of the peripheral insulin resistance occurring in T2D [44]. In obesity, increased and sustained concentrations of fatty acids in the peripheral bloodstream contribute to establishing lipotoxic stimuli and a chronic inflammatory state impairing ER stress [45]. A clear consequence of these homeostatic alterations is the activation of the unfolded protein response (UPR), commonly upregulated in cancer [46].

A critical revision of the scientific literature, concerning the influence of food ingredients on NOD/NLR expression, shows their potential to partially, either with a down-/up-regulatory effect, affect NODs/NLRs [42,47]. In addition, both Gram (+) and (−) bacteria release extracellular vesicles, but knowledge of the genus *Lactobacillus* and *Bifidobacterium* remains poor and comprehensive and integrated knowledge is needed to understand their capacity to activate/control NOD signaling. Similarly, compelling revisions can be found regarding the potential of naturally occurring bioactive compounds (i.e., mostly polyphenols and associated molecules) displaying inhibitory activity in regard to protein kinases sensed by NODs [48]. However, a common gap in knowledge is that the exact mechanism remains to be elucidated. These aspects limit the attribution of added value to food ingredients as modulators of the NOD/NLR axis with a preventive or therapeutic intention.

## 4. Immunonutritional Interventions on NOD Signaling

The innate immune system is deeply and complexly linked not only to immune, but also to metabolic homeostasis, and thus associated with different diseases [49]. Despite preclinical mouse models having provided a way to study human innate immune pathways, many cases lacking translation to humans can be found, which is most likely attributable to developmental and evolutionary aspects [50]. The divergence of NODs between humans and mice appears to not be so extensive in the inflammasome-forming NLRP and IPAF subfamilies [51]. Otherwise, different genomic content in the TLRs as well as regulated response(s) in mouse and human macrophages has been reported. Thus, immune and metabolic processes in NCDs [52,53,54,55] and cancer [56] can be exacerbated during disease development. These differences can have significant implications for devising the potential of immunonutritional strategies to influence NODs.

The scientific literature reveals scarce information from patients concerning the involvement of specific NODs in NCDs [57,58]. NOD1/2 were shown to be up-regulated (mRNA) in monocytes from patients with T2D. This result allows us to hypothesize that nutrient utilization by monocytes is going to be affected in such a way that insulin resistance is increased thereby signaling events that promote an inflammatory phenotype [59]. Synergistic effects between NOD-like receptors (NLRs) and TLRs in human B lymphocytes [60] have been reported. Thus, immunometabolic adaptations in the myeloid population promote a cascade of events involving adaptive T-cells [61] leading to increased, ultimately aberrant and uncontrolled, inflammatory processes. In obese individuals, it was shown that NLRP3 activation could be attenuated by acutely raising the plasmatic concentration of β-hydroxybutyrate with the ingestion of exogenous ketones [58]. This effect was reflected in the decreased production of plasmatic IL-1β. Together with IL-1α, these cytokines significantly contribute to the promotion of insulin resistance impairing the function of adipocytes in promoting inflammation. In addition to low-grade inflammation, NCDs [62] share other features such as imbalances in the composition of the gut microbiota, which have also been associated with cancer [63]. In this sense, patients suffering from intestinal metaplasia display a modified cluster of microbes after *H. pylori* eradication [64]. Despite the abundance of pathogenic bacteria and the presence of oral microbes enabling the production of peptidoglycan(s), decreased NOD-like signaling was found. This effect appears to contrast with that expected from the endotoxin-like properties of the peptidoglycan(s) from pathogens. The microbial cluster of these patients was composed of, among others, members of the *Prevotella*, *Rothia*, and *Granulicatella* genera, which could represent a good source of butyrate-producing bacteria [65,66]. It is worth noting the butyrate-induced effects limiting natural killer (NK) cell function [67], which could allow hypothesizing this effect to explain, at least in part, the reported data on patients suffering intestinal metaplasia. It is known that NK cells play a dual role, and in cases of immune evasion even help promote metastases [68]. This hypothesis could also be supported by the anti-inflammatory effects derived from probiotic administration. Despite the widespread use of probiotics, most of which include the *Bifidobacterium* and *Lactobacillus* genera producing peptidoglycans, there is scarce information from controlled trials concerning their direct association either with the expression or modulation of NODs and NOD-like receptors (NLRs). In this regard, we can hypothesize that the production of peptidoglycan hydrolases by these genera could present a different activity than those produced by Gram (−) bacteria. Thus, despite the generation of ligands for NOD2 by Gram (+) bacteria [69], the signaling of this receptor results in effects that may not synergize with TLRs, which would partly explain the absence of inflammatory effects (Figure 2). Otherwise, a significant bulk of scientific reports exists confirming that cytokine production and innate immune as well as adaptive cells are modulated by lactobacilli and bifidobacteria resulting in improved regulatory immune responses [70].

Currently, most of the prominent information concerning the key role of NODs and NLRs in immunity and metabolism has come from preclinical models [71,72]. In addition to identifying the different signaling pathways and interaction with PAMPs and DAMPs, studies have demonstrated the enormous involvement of these receptors in different immunonutritional-based processes and diseases, including cancer [73,74,75]. These studies constitute good examples of the many immune and metabolic diseases associated with NODs and NLRs. Prior research defined the protective role of NOD1/2 deficiency from HFD-induced inflammation, lipid accumulation, and peripheral insulin intolerance [20]. Further studies also defined an active role for NOD1 in the inflammatory environment associated with both experimental and human diabetic cardiac disease [76]. In addition, recent data point out the protective function of NOD1 reducing low-grade inflammation and thereby obesity development [21]. Alterations in glucose metabolism have been previously identified as a major driver of B cell lymphopoiesis and function [77]. Recently, a key—microbial ligand-independent—role has been demonstrated for NOD1in regulating proliferative response(s) in adaptive immunity [75]. Notably, the latter response(s) were associated with NOD1-mediated contrasting effects preventing colitis, while impairing T cell maturation and activity. Thus, important consequences during anti-tumoral immunity development can be expected. A clear example is the recently described microbiota-dependent activation of CD4^+^ T cells inducing the CTLA-4 blockade via Fcγ receptors [78]. However, the *Bifidobacterium* and *Lactobacillus* genera prevent immunological alterations, generally causing severe inflammation, associated with the activation of CTLA4 [79,80]. Supplementation with probiotics can help to integrate the diverse biological response(s) derived from NOD/NLR activation, PAMPs, and DAMPs, the lack of activation by their peptidoglycan(s), but control of the proinflammatory milieu and TLR signaling (Figure 2). This speculation could be supported by in vivo studies using a probiotic mixture (i.e., *Bifidobacterium* strains and *Lacticaseibacillus rhamnosus*) as a supplement for preterm infants, where it was proven effective in reducing the level of calprotectin as well as IFN-γ and IL-22 [81], the latter mediating the synergic and inhibitory effects, respectively, of NOD activity and expression. The administration of prebiotic inulin, which increased the abundances of *Akkermansia* and *Bifidobacterium* that was reflected in a decrease in the ratio *Firmicutes/Bacteroidetes,* was shown to cause the down-regulation of NLRP3 [82]. Also, some other symbionts such as *Bacterioides fragilis*, *Enterococcus fecalis*, and *Lactobacillus plantarum* have been identified as microbes that do not induce the NLRC4-dependent release of IL-1β [83].

Obesity and T2D result in enhanced oxidative damage to circulating lipoproteins in the plasma, playing a key role in the pathogenesis of the diseases. It is not surprising that molecular components of LDL such as phospholipids as well as parts of the cell membrane become oxidized (oxPLs), thus exerting critical functions as immunomodulatory (DAMPs) signals [84]. These oxPLs can interact with CD14 and be recognized by CD36 accessing the cytosol to activate caspases 1 and 11, the NLRP3 regulators [85]. In addition, in vitro studies have also shown the role of oxPLs in inducing the chemotaxis and intracellular calcium influx in natural killer (NKs) cells [86] and the release of IL-6 in human monocytes [87]. It is worth noting here that NOD-like receptors may directly mediate signaling at the endoplasmic reticulum affecting calcium influx [34]. These signals end up in the development of an inflammatory milieu and could drive the hyperactivation of innate lymphoid cells (ILCs) group 1, which includes the conventional and unconventional subsets of NK cells. Notably, ILCs aggravate adipose tissue fibrosis and the development of diabetes in obesity [88].

From an immunonutritional perspective, n-3 PUFAs represent important food ingredients with the potential to improve, at least in part, NCD-associated immune and metabolic imbalances alleviating obesity, T2D, and metabolic syndrome, and they could also slow down tumor growth and increase the efficacy of chemotherapy. In particular, n-3 PUFAs have been shown to be effective in inhibiting NF-kappaB and IL-8 expression induced by NOD receptors in HCT116 [89]. So far, the results derived from the administration of n-3 PUFAs in humans on the expansion and activation of the immunological effectors of the innate branch—NKs and macrophages—are not conclusive. The scarce literature on the matter reveals the variability of biological responses and their meaning [90,91,92,93]. The consumption of n-3 PUFAs had a significant influence on the functional polarization of the macrophage population in adipose tissue, promoting a drift toward its M2 subphenotype while reducing insulin resistance [93]. Recent research suggests that the body’s composition in PUFAs appears to be associated with TLR4 signaling [16,17,18,19,20,21,22,23,24,25,26,27,28,29,30,31,32,33,34,35,36,37,38,39,40,41,42,43,44,45,46,47,48,49,50,51,52,53,54,55,56,57,58,59,60,61,62,63,64,65,66,67,68,69,70,71,72,73,74,75,76,77,78,79,80,81,82,83,84,85,86,87,88,89,90,91,92,93,94]. The latter is attributed to a protein fraction (~30 kDa MWCO) isolated from *C. quinoa* with advantageous effects, either in healthy or cancer-developing mice—improving mechanisms to control inflammatory processes and liver macrophage and ILC expansion in animals under HFD. These results allow us to hypothesize that the derivation of signaling in TLRs can condition and modify the severity and orientation of NOD control over metabolic and immunological responses.

## 5. Conclusions and Future Perspectives

While significant progress has been made to better understand the role of NODs/NLRs in human diseases, immunity, and inflammation, the relation to NCDs and the interaction with immunonutritional compounds remain poorly defined from a biomedical point of view. A critical review of the literature clearly shows the scarcity of existing information, and the non-specific effects of food/food-derived ingredients in interactions with NOD/NLR receptors. Lacking ‘defined’ and ‘oriented mechanism(s)’ limits the use of food with a preventive and/or therapeutic intention further than considering their nutritional profile. It remains to be understood how these receptors differentiate between peptidoglycan from pathogenic bacteria and beneficial probiotics. In addition, how NODs/NLRs integrate the stimuli and TLR-derived signaling enabled by immunonutritional compounds needs to be elucidated. Altogether, only in conjunction with the dynamics of immune and metabolic alterations occurring at NCDs can the role, worsening or improving disease, be elucidated. A better understanding of their molecular implication and influence on innate immune effector cells—myeloid monocytes/macrophages and the expansion of ILCs—can significantly contribute to control and/or intervention in NCDs and cancer (Figure 1). The use of biocompatible vesicles could help in improving the bioavailability of biofunctional food compounds to influence the activity of NODs/NLRs.

NCDs and cancer could benefit from early and preventive immunonutritional strategies, which result in less aggressive to physiological processes. Although NODs/NLRs have been considered in various clinical interventions, improved research into immunonutritional trials is needed along with large-scale studies. The integration of this knowledge and the definition of the lack of understanding will significantly contribute to paving the way to transdisciplinary interventions, enabling nutrition precision interventions with a preventive or therapeutic intention in regard to immune and metabolic imbalances leading to NCDs and cancer. Understanding the role of NODs/NLRs may represent an important molecular checkpoint for devising effective strategies for the innate immune control of these diseases.

## Figures and Tables

**Figure 1 biomedicines-12-00341-f001:**
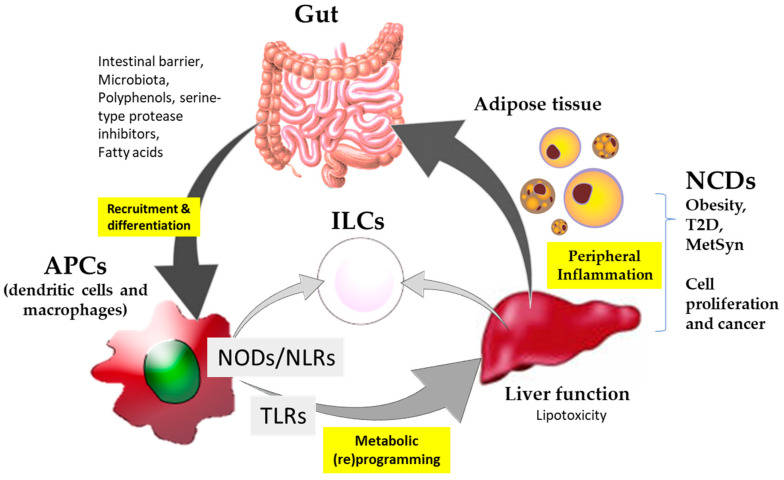
Schematic summary of the participation of immunonutritional agonists in interaction with the nucleotide-binding oligomerization domain-like (NODs) and ‘Toll-like’ (TLRs) receptors and their implication for non-communicable diseases (NCDs) and cancer.

**Figure 2 biomedicines-12-00341-f002:**
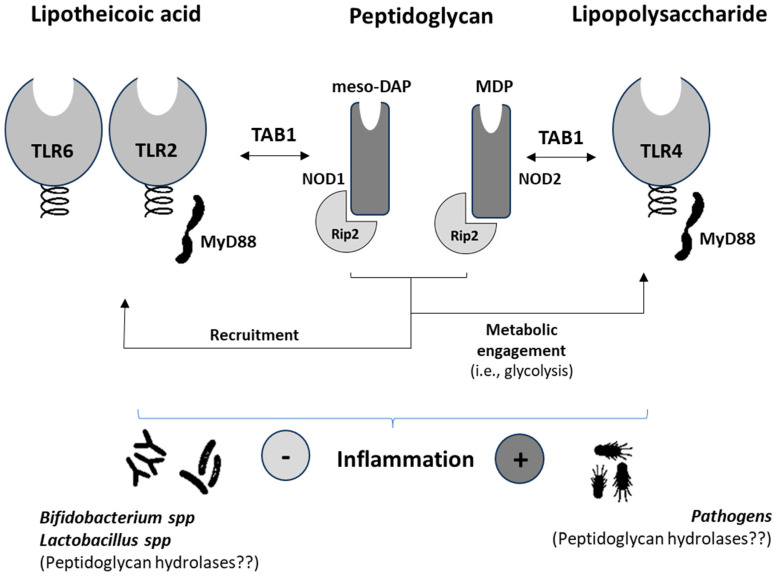
Schematic diagram summarizing the synergies between nucleotide-binding oligomerization domain-containing proteins (NOD) and Toll-like receptors in human immune and metabolic systems and their main agonists. meso-DAP, γ-D-glutamyl-meso-diaminopimelic acid; MDP, muramyl dipeptide; TAB1, mitogen-activated protein kinase kinase 7-interacting protein-1; Rip2, receptor-interacting protein 2.

## Data Availability

Not applicable.

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
