# Peer review of "Contribution of Nucleotide-Binding Oligomerization Domain-like (NOD) Receptors to the Immune and Metabolic Health"

_biomedicines, 2024, doi:10.3390/biomedicines12020341_

Round 1

Reviewer 1 Report

Comments and Suggestions for Authors

Comments to the Authors of manuscript number: biomedicines-2841290 entitled “Contribution of NODs to the immune and metabolic health”.

 his review provides insights into the complex relationship between dietary factors and NOD1/2, shaping immune and metabolic health.

1. Introduction:

a. The introduction lacks a clear and concise statement of the research question or objective. Readers should be informed early on about the specific focus or aim of the review. Consider restructuring the introduction to provide a smoother flow of information, guiding the reader through the rationale and significance of the study.

b. While statistics on global obesity, hyperglycemia, and cancer are provided, the source of these data is not explicitly mentioned. Citing the specific references for these statistics would enhance the credibility of the information.

c. Some information is repeated, such as the mention of the impact of metabolic diseases like obesity and diabetes. Redundancies should be minimized to maintain reader engagement.

d. The introduction touches upon various aspects, from global health statistics to the role of immune components and nutritional interventions. Consider narrowing the focus and delving into specific aspects gradually to maintain clarity.

e. abbreviations such as NCDs, IFs, and NODs might be overwhelming for readers initially. Consider introducing and explaining these abbreviations when first used.

f. Some sentences are complex, making the text challenging to follow. Aim for clarity and simplicity in conveying complex scientific concepts.

2. Current challenges in the nutritional approach of NODs in NCDs and cancer:

a. The part on current challenges in the nutritional approach of NODs in NCDs and cancer provides valuable insights.

b. The text would benefit from clearer articulation of the challenges faced in the nutritional approach to NCDs and cancer. Define specific gaps or limitations in current approaches.

c. Acknowledge opposing views or conflicting results in preclinical studies, such as the varied roles of NOD1 in obesity. This adds nuance to the discussion and acknowledges the complexity of the topic.

d. Highlight areas that remain poorly explored, particularly regarding the modulation of immune effectors by foods/food constituents. This emphasizes the need for further research in specific aspects.

e. Clarify hypotheses, such as the assumed inhibitory regulation of NOD1 in the context of NCDs. Provide a rationale for these assumptions and acknowledge uncertainties in the current understanding.

f. Ensure consistent use of terminology. For example, the use of "precision nutrition intervention with an immunonutritional intention" could be simplified for better comprehension without losing the core concept.

g. While emphasizing the potential benefits of precision nutrition, maintain a balanced view by acknowledging potential challenges or limitations associated with this approach.

h. Discuss the potential practical implications of the findings. How can the identified challenges inform future research or guide nutritional interventions in a clinical setting?

i. While preclinical studies are valuable, consider discussing the relevance of these findings to human subjects. Highlight any existing or lacking evidence from human studies.

3. NODs/NLRs signaling: implications for NCDs and cancer.

a. Acknowledge any controversies or differing opinions in the scientific community regarding the role of NOD1/2 in specific diseases.

b. Reinforce the connection between NODs, Toll-like receptors (TLRs), and the overall theme of their contribution to immune and metabolic imbalances in NCDs and cancer. Clearly articulate how these receptors contribute to disease development.

c. When discussing gaps in knowledge, go beyond stating that mechanisms remain to be elucidated. Engage in a critical evaluation of these gaps and discuss the implications for future research or potential challenges in harnessing NODs for preventive or therapeutic purposes.

d. Discuss the potential practical implications of the findings. How can the knowledge about NODs inform dietary recommendations or therapeutic interventions?

e. Explicitly discuss the relevance of the findings to human health. How do the molecular processes involving NODs translate to potential interventions or treatments for individuals with NCDs or cancer?

 4. Immunonutritional interventions on NOD signaling

a. Emphasize the relevance of the discussed findings to human health. Explicitly connect the observed effects in preclinical models to potential implications for human disease and immunonutritional interventions.

b. When discussing gaps in knowledge, consider proposing potential avenues for future research.

c. Reinforce how the discussed immunonutritional interventions relate to the main thesis of the passage, particularly regarding NODs and their role in immune and metabolic imbalances.

d. Discuss the practical implications of the findings. How can the insights into NOD signaling inform dietary recommendations or therapeutic strategies for individuals with NCDs or cancer?

5. conclusion and perspectives:

a. Clarify the biomedical context by specifying which NCDs and types of cancer are particularly relevant to the discussion.

b. Emphasize the importance of addressing the remaining questions and uncertainties.

c. Suggest practical steps for future research.

d. Clearly articulate how a deeper understanding of NODs/NLRs can directly translate into clinical interventions.

e. Discuss how findings in this area could lead to tangible advancements in preventive and therapeutic strategies.

f. Stress the need for interdisciplinary collaboration between nutrition, immunology, and clinical research to fully unlock the potential of immunonutritional strategies.

 6. there is no part describing the literature search, usually presented in the form of material and methods.

Author Response

Rev_1

Comments and Suggestions for Authors

Comments to the Authors of manuscript number: biomedicines-2841290 entitled “Contribution of NODs to the immune and metabolic health”. This review provides insights into the complex relationship between dietary factors and NOD1/2, shaping immune and metabolic health.

  1. Introduction:
  2. The introduction lacks a clear and concise statement of the research question or objective. Readers should be informed early on about the specific focus or aim of the review. Consider restructuring the introduction to provide a smoother flow of information, guiding the reader through the rationale and significance of the study.

According to the reviewer’s comment the introduction has been revised throughout to clarify the text and improve its understanding.

  1. While statistics on global obesity, hyperglycemia, and cancer are provided, the source of these data is not explicitly mentioned. Citing the specific references for these statistics would enhance the credibility of the information.

Many thanks for your comment, which we really appreciate. The reason why we did not include any reference is because these statistics are very general and the World Health Organization is well known.

According to the reviewer’s comment, it has been indicated the link (https://www.who.int/) to the webpage of the WHO where a complete information of these pathologies can be found.

  1. Some information is repeated, such as the mention of the impact of metabolic diseases like obesity and diabetes. Redundancies should be minimized to maintain reader engagement.

We feel sorry for this misunderstanding. The introduction section describes different aspects concerning the relevance of chronic metabolic diseases, establishes their social impact, and defines the role of innate immune effectors in these diseases. From our perspective this organization is needed to help the reader understand the context and potential importance of food-mediated modulation of NODs.

  1. The introduction touches upon various aspects, from global health statistics to the role of immune components and nutritional interventions. Consider narrowing the focus and delving into specific aspects gradually to maintain clarity.

The manuscript aims to ‘…help to integrate the potential use of NODs in targeted precision nutrition strategies as coadjuvant to the classical pharmacological treatment of NCDs and cancer.’ (page 3, line 110-111). From our perspective a review must expose the whole context and help the transit of readers from the general context to the specific molecular interest. The latter constitutes the main motivation to include various aspects in the introduction. Afterwards, along the manuscript are developed the molecular approaches and potential consequences of modulating the activity of NODs receptors.

Notably, the manuscript reveals the scarce information existing with this respect (page 7, line 277-278). Obesity, T2D, metabolic syndrome, NAFLD and cancer are heterogeneous and multifaceted diseases. This makes necessary to provide a holistic (as much as possible) view of these to help the reader understand the potential relevance of the strategies approaching the diseases. We have worked to comprehensively structure the relevance of the diseases, exploring their connection with the components of the immune system and how, in turn, these are influenced by nutrients.

  1. abbreviations such as NCDs, IFs, and NODs might be overwhelming for readers initially. Consider introducing and explaining these abbreviations when first used.

In view of the reviewer’s comment, the abstract and introduction has been rewritten to introduce the meaning of these abbreviations.

  1. Some sentences are complex, making the text challenging to follow. Aim for clarity and simplicity in conveying complex scientific concepts.

In the light of the reviewer’s comment the introduction has been revised throughout to clarify the text and improve its understanding.

  1. Current challenges in the nutritional approach of NODs in NCDs and cancer:
  2. The part on current challenges in the nutritional approach of NODs in NCDs and cancer provides valuable insights.

Many thanks for your comment which we really appreciate.

  1. The text would benefit from clearer articulation of the challenges faced in the nutritional approach to NCDs and cancer. Define specific gaps or limitations in current approaches.

Many thanks for your comment. According to the reviewer’s comment specific knowledge unmet gaps have been described in the text (Page 5, line 178-185)

  1. Acknowledge opposing views or conflicting results in preclinical studies, such as the varied roles of NOD1 in obesity. This adds nuance to the discussion and acknowledges the complexity of the topic.

Many thanks for your comment which we really appreciate.

  1. Highlight areas that remain poorly explored, particularly regarding the modulation of immune effectors by foods/food constituents. This emphasizes the need for further research in specific aspects.

According to the reviewer’s comment this need has been defined as a current unmet gap in the manuscript (Page 5, line 178-185).

  1. Clarify hypotheses, such as the assumed inhibitory regulation of NOD1 in the context of NCDs. Provide a rationale for these assumptions and acknowledge uncertainties in the current understanding.

We really appreciate your comment, which we did not want to make it too much speculative. The rationale to assume an inhibition of NODs in to agree with the ‘…reduced macrophage apoptosis derived from NOD1 deficiency [21]…’ (page 4, line 148-157). NOD1 deficiency reduces macrophage apoptosis, thus as it is found an increased proportion of macrophages in NCDs, we suggest that, somehow, there should be some inhibitory factors on these receptors.

In the light of the reviewer’s comment the word ‘hypothesis’ has been rewritten as ‘consider’ (page 4, line 154).

  1. Ensure consistent use of terminology. For example, the use of "precision nutrition intervention with an immunonutritional intention" could be simplified for better comprehension without losing the core concept.

According to the reviewer’s comment, this terminology has been rewritten (page 5, line 178-183).

  1. While emphasizing the potential benefits of precision nutrition, maintain a balanced view by acknowledging potential challenges or limitations associated with this approach.

We fully agree with the reviewer’s comment. The text already states that precision nutrition should be ‘…complementary…’ (page 5, line 186).

In view of the reviewer’s comment, some sentences in the conclusion section have also been rewritten as follows ‘…could benefit from early and preventive immunonutritional strategies, which result less aggressive to physiological processes.’ (page 10, lines 419-421).

  1. Discuss the potential practical implications of the findings. How can the identified challenges inform future research or guide nutritional interventions in a clinical setting?

According to the reviewer’s comment, current knowledge unmet gaps have been specified in the text (page 5, line 178-183). In a clinical setting, ‘econutritional conditions’ as well as ‘understanding’ the ‘specific’ effects of food-derived ingredients must be defined.

  1. While preclinical studies are valuable, consider discussing the relevance of these findings to human subjects. Highlight any existing or lacking evidence from human studies.

The main benefit for humans would be a better control of the immune and metabolic conditions. The latter would improve the nutrient utilization minimizing lipotoxic effects, peripheral inflammation and cellular proliferation (Figure 1). As stated in the text, these effects could be derived from a better control, among others, on insulin sensitivity, ER stress, RhoGTPases, etc (page 6).

  1. NODs/NLRs signaling: implications for NCDs and cancer.
  2. Acknowledge any controversies or differing opinions in the scientific community regarding the role of NOD1/2 in specific diseases.

NOD-like receptors have been described as master regulators of innate immunity. In general terms, the whole scientific community agrees on the roles for NODs in ER stress-driven chronic inflammatory diseases such as obesity, and Type 2 diabetes. However, to date, it is a compartmentalized approach to the modulation of these receptors. Here, the nutritional community did not receive too much attention and motivated this review. Move further than already discussed in the document is limited as stated in the manuscript, there is no ‘direct’ information concerning the specific pathways used by food-ingredients to modify the expression of NODs.

  1. Reinforce the connection between NODs, Toll-like receptors (TLRs), and the overall theme of their contribution to immune and metabolic imbalances in NCDs and cancer. Clearly articulate how these receptors contribute to disease development.

We feel sorry for the misunderstanding as the relationship between those is already stated in the manuscript (page 5, line 212-214). NCDs are characterized by low-grade inflammation. In this context, peripheral monocytes and macrophages derived from these appear as central players in the progression of liver dysfunction, worsening or improving the disease (page 4, line 153-155). Nucleotide-binding oligomerization domain-like (NOD) receptors rely on the interface between immunity and metabolism (abstract). Thus, all parameters involved in developing alterations in the immune and metabolic function is critical and a potential target to approach these diseases. A clear articulation of the NODs contribution is impossible as many aspects remain ill-defined.

  1. When discussing gaps in knowledge, go beyond stating that mechanisms remain to be elucidated. Engage in a critical evaluation of these gaps and discuss the implications for future research or potential challenges in harnessing NODs for preventive or therapeutic purposes.

We fully agree with the reviewer that gaps in knowledge should be fulfilled. It is not possible to discuss more, without moving into too much speculative opinions, as we cannot advance more effects rather than controlling immune differentiation and function and metabolic control (figure 1). Overall, it is agreed the proinflammatory role of NODs; however, probiotics can induce NODs activation but only cause anti-inflammatory effects (figure 2). These questions, already discussed in the manuscript (page 9, line 336).

  1. Discuss the potential practical implications of the findings. How can the knowledge about NODs inform dietary recommendations or therapeutic interventions?

As NODs rely on the interface between immunity and metabolism, a better understanding of the role that food-ingredients can play as signaling regulators will have a direct impact on nutritional recommendations in establishing precision nutrition strategies.

  1. Explicitly discuss the relevance of the findings to human health. How do the molecular processes involving NODs translate to potential interventions or treatments for individuals with NCDs or cancer?

We feel sorry for this misunderstanding. As already state din the manuscript, ‘…NOD1/2 activation and deficiency as well as the intake of a diet rich in lipids and certain signals of cellular damage are closely related to cell proliferation and the response(s) to chemotherapy in hepatocellular carcinoma (HCC) [31]’ (page 6, line 220-222). See previous comment.

  1. Immunonutritional interventions on NOD signaling
  2. Emphasize the relevance of the discussed findings to human health. Explicitly connect the observed effects in preclinical models to potential implications for human disease and immunonutritional interventions.

Many thanks for your comment. In our opinion, these aspects are answered in the previous comments.

  1. When discussing gaps in knowledge, consider proposing potential avenues for future research.

Many thanks for your comment. According to the reviewer’s comment specific knowledge unmet gaps have been described in the text (Page 5, line 178-185)

  1. Reinforce how the discussed immunonutritional interventions relate to the main thesis of the passage, particularly regarding NODs and their role in immune and metabolic imbalances.

We feel sorry for the misunderstanding. NCDs are characterized by low-grade inflammation. In this context, peripheral monocytes and macrophages derived from these appear as central players in the progression of liver dysfunction, worsening or improving the disease (page 4, line 153-155). Nucleotide-binding oligomerization domain-like (NOD) receptors rely on the interface between immunity and metabolism (abstract). Thus, all parameters involved in developing alterations in the immune and metabolic function is critical and a potential target to approach these diseases. A clear articulation of the NODs contribution is impossible as many aspects remain ill-defined.

  1. Discuss the practical implications of the findings. How can the insights into NOD signaling inform dietary recommendations or therapeutic strategies for individuals with NCDs or cancer?

We feel sorry, but this question seems to be duplicate. As NODs rely on the interface between immunity and metabolism, a better understanding of the role that food-ingredients can play as signaling regulators will have a direct impact on nutritional recommendations in establishing precision nutrition strategies.

  1. conclusion and perspectives:
  2. Clarify the biomedical context by specifying which NCDs and types of cancer are particularly relevant to the discussion.

NCDs such as obesity and T2D are closely associated to, among other, colorectal cancer (Aliment Pharmacol Ther 2022, 56(3):407-418; JAMA Netw Open. 2023;6(11):e2343333), hepatocarcinoma (Biochim Biophys Acta Rev Cancer 2018, 1869(2):97-102; Biomed Res Int. 2017; 2017: 5202684). This information is well-known, and that motivated our reference to cancer in general terms). Without information of what food can modulate NODs? and what direction? Describe the biomedical context of these pathologies wouldn’t provide additional information as econutritional conditions can be completely opposite.

  1. Emphasize the importance of addressing the remaining questions and uncertainties.

Many thanks for your comment. According to the reviewer’s comment specific knowledge unmet gaps have been described in the text (Page 5, line 178-185). The importance of addressing these questions is supported by the enormous impact of NCDs on the society.

  1. Suggest practical steps for future research.

We really appreciate your comment. Practical steps for future are described in the knowledge unmet gaps.

  1. Clearly articulate how a deeper understanding of NODs/NLRs can directly translate into clinical interventions.
  2. Discuss how findings in this area could lead to tangible advancements in preventive and therapeutic strategies.

We feel sorry, but this question seems to be duplicate. As NODs rely on the interface between immunity and metabolism, a better understanding of the role that food-ingredients can play as signaling regulators will have a direct impact on nutritional recommendations in establishing precision nutrition strategies. To this end it is needed to understand what food ingredients? And how NODs’ signaling is modified?.

  1. Stress the need for interdisciplinary collaboration between nutrition, immunology, and clinical research to fully unlock the potential of immunonutritional strategies.

A clear example of the benefits of these collaborations are the studies described in the references Biomed 2021, 9, 1633, doi: 10.3390/biomedicines9111633 and Foods 2023, 12, 3321, doi: 10.3390/foods12173321. These are already included in the manuscript.

  1. there is no part describing the literature search, usually presented in the form of material and methods.

We really appreciate your comment. Usually, description of the literature search is cincluded in systematic reviews, which is not the case of this document.

According to the reviewer’s comment a brief sentence has been included in the manuscript (page 3, line 123-129).

Reviewer 2 Report

Comments and Suggestions for Authors

Dear Authors

Thank you for your manuscript submission. The topic of the manuscript is very interesting. However, the main text suffers from the lack of the following items:

1. This manuscript suffers from the lack of the related signaling pathways. There are several signaling pathways that are involved. Please do add a subtitle in this regard.

2. This manuscript suffers from the lack of the main mediators and modulators in the related signaling pathways.

3. It is recommended to add some figures to show the main signaling pathways in this regard.

4. It is recommended to add a table to show the main mediators, modulators, etc., their structures, their pivotal roles and their effects on the pathways.

5. The SNPs have determined role in the field. A subtitle should be added in this regard.

6. It is recommended to present the importance of agonists and antagonists in the field.

7. It is recommended to read and add the following papers to References section to have a fruitful manuscript:

Toll-like receptor-guided therapeutic intervention of human cancers: molecular and immunological perspectives. Front Immunol. 2023 Sep 26;14:1244345. doi: 10.3389/fimmu.2023.1244345. PMID: 37822929; PMCID: PMC10562563.

The Interleukin-1 (IL-1) Superfamily Cytokines and Their Single Nucleotide Polymorphisms (SNPs). J Immunol Res. 2022 Mar 26;2022:2054431. doi: 10.1155/2022/2054431. PMID: 35378905; PMCID: PMC8976653.

  Comments on the Quality of English Language

Moderate Edition is needed

Author Response

Rev_2

Comments and Suggestions for Authors

Dear Authors

Thank you for your manuscript submission. The topic of the manuscript is very interesting. However, the main text suffers from the lack of the following items:

Many thanks for your comment, which we fully join. A critical review of the literature reveals the scarce studies existing in this field. Notably, most of the existing studies only display the unspecific effects of foods on NODs/NLRs signaling. Food/food-derived ingredients appear as signaling modulators; however, without a ‘defined’ and ‘oriented’ fashion the use of those with a clinical preventive and/or therapeutic intention appears limited.

  1. This manuscript suffers from the lack of the related signaling pathways. There are several signaling pathways that are involved. Please do add a subtitle in this regard.

Many thanks for your comment, which we really appreciate. The fact that mechanism(s) remain undefined is the motivation of this document (Page 6, line 246-248). Most studies in the literature report the influence of food/food-derived ingredients on NODs/NLrs expression and/or inhibition. However, these unspecific effects do not enable the use of food in a preventive/therapeutic intention. Our intention is to highlight the lack of knowledge and definition of the underlying molecular mechanisms, despite probiotics can produce agonists of NODs receptors. Why does that occur? We propose some hypothesis, to highlight the need of knowledge in some areas.

It is very difficult to add a subtitle in this line when there is a lack of definition in the molecular mechanism(s) (Page 6, line 246-248)

  1. 2. This manuscript suffers from the lack of the main mediators and modulators in the related signaling pathways.

The manuscript focuses on the food/food-derived ingredients to modulate molecular signaling of NODs/NLRs. Particularly, the review aims to clarify the specific mechanisms(s) engaged by food/food-derived ingredients. A critical review of the literature shows the unspecific effects and the lack of definition on the mechanisms triggered by these compounds. This document tries to reveal the importance of the NOD/NLRs receptors in NCDs, but highlighting the lack of data in this sense.

It is known the synergic effects between TLRs and NODs, but there is a lack of data concerning the modulatory role of specific food ingredients in molecular signaling, direct interaction, to orient and modify molecular signaling to be used in a therapeutic intention.

  1. It is recommended to add some figures to show the main signaling pathways in this regard.

We feel sorry, but as the molecular mechanism(s) have not been defined it is difficult to show a concrete mechanism. The existing knowledge is summarized in Figure 2, but more defined interaction would imply too much speculation as the knowledge is not defined.

  1. It is recommended to add a table to show the main mediators, modulators, etc., their structures, their pivotal roles and their effects on the pathways.

We feel sorry for this misunderstanding. Figure 2 summarizes the current knowledge in this sense.

  1. The SNPs have determined role in the field. A subtitle should be added in this regard.

To the best of our knowledge there is a lack concerning the food-derived ingredients to interact with NOD/NLRs receptors not the SNPs determining the inflammatory response(s)

  1. It is recommended to present the importance of agonists and antagonists in the field.

This study focuses on the food-derived ingredients enabling the NODs/NLRs interaction, which reveals the lack of knowledge to explain the probiotic signaling.

  1. It is recommended to read and add the following papers to References section to have a fruitful manuscript:

Toll-like receptor-guided therapeutic intervention of human cancers: molecular and immunological perspectives. Front Immunol. 2023 Sep 26;14:1244345. doi: 10.3389/fimmu.2023.1244345. PMID: 37822929; PMCID: PMC10562563.

The Interleukin-1 (IL-1) Superfamily Cytokines and Their Single Nucleotide Polymorphisms (SNPs). J Immunol Res. 2022 Mar 26;2022:2054431. doi: 10.1155/2022/2054431. PMID: 35378905; PMCID: PMC8976653.

We fully agree on the importance of the aspects approached in the proposed articles. The present document focuses on the context of NODs modulation by food-ingredients. The proposed articles, despite providing very important information are far from the objective of this study.

Reviewer 3 Report

Comments and Suggestions for Authors

The manuscript entitled "Contribution of NODs to the immune and metabolic health" should be substantially revised before publication.

Title must not contain abbreviations.

References on the role of phytochemicals and extracts of aromatic and medicinal plants, such as oregano, thyme, sage or others are totally missing. Rereferences on probiotics and in general on feed additives or feed constituents, their role and impact on gene expression or metabolism regulation are totally missing. Conclusions, should be revised according to the new information that is a necessary requirement. 

Author Response

Rev_3

Comments and Suggestions for Authors

The manuscript entitled "Contribution of NODs to the immune and metabolic health" should be substantially revised before publication.

Title must not contain abbreviations.

In view of the reviewer’s comment the title has been rewritten as follows ‘Contribution of Nucleotide-binding oligomerization domain (NODs) receptors to the immune and metabolic health

References on the role of phytochemicals and extracts of aromatic and medicinal plants, such as oregano, thyme, sage or others are totally missing. References on probiotics and in general on feed additives or feed constituents, their role and impact on gene expression or metabolism regulation are totally missing.

Many thanks for your comment. This document aims to discuss the existing literature displaying the potential of food ingredients to interact with the NODs/NLRs receptors. Importantly, our intention has been to display the information that can be associated to a ‘defined’ and ‘oriented’ mechanism(s). A critical review of the literature clearly shows the scarce existing information, and the unspecific effects of those. Lacking ‘defined’ and ‘oriented mechanism(s) disables the use of food with a preventive and/or therapeutic intention further than considering their nutritional profile.

In view of the reviewer’s comment, these aspects have been clarified in the introduction section. Besides, few additional references have been included to cover the reviewer’s concern (page 4, line 168 – page 5, line 175).

Dou X, Yan D, Liu S, Gao L, Shan A. Thymol Alleviates LPS-Induced Liver Inflammation and Apoptosis by Inhibiting NLRP3 Inflammasome Activation and the AMPK-mTOR-Autophagy Pathway. Nutrients. 2022, 14(14):2809. Doi: 10.3390/nu14142809.

Choudhari AS, Mandave PC, Deshpande M, Ranjekar P, Prakash O. Phytochemicals in Cancer Treatment: From Preclinical Studies to Clinical Practice. Front Pharmacol. 2020, 10:1614. doi: 10.3389/fphar.2019.01614. Erratum in: Front Pharmacol. 2020 Feb 28;11:175.

PÅ‚óciennikowska A, Hromada-Judycka A, BorzÄ™cka K, Kwiatkowska K. Co-operation of TLR4 and raft proteins in LPS-induced pro-inflammatory signaling. Cell Mol Life Sci. 2015 Feb;72(3):557-581. doi: 10.1007/s00018-014-1762-5.

Conclusions, should be revised according to the new information that is a necessary requirement.

The text has been revised accordingly to the new information included in the manuscript (page 9, line 382-386).

Round 2

Reviewer 2 Report

Comments and Suggestions for Authors

Accept

Reviewer 3 Report

Comments and Suggestions for Authors

Authors have revised their work substantially and in a satisfactory way.